

# A multi-model approach to monitor emissions of $CO_2$ and CO from an urban-industrial complex

Ingrid Super[1,2], Hugo A.C. Denier van der Gon[2], Michiel K. van der Molen[1], Hendrika A.M. Sterk[3], Arjan Hensen[4], Wouter Peters[1,5]

[1] Meteorology and Air Quality Group, Wageningen University, P.O. Box 47, 6700 AA Wageningen, Netherlands
[2] Department of Climate, Air and Sustainability, TNO, P.O. Box 80015, 3508 TA Utrecht, Netherlands
[3] National Institute for Public Health and the Environment, P.O. Box 1, 3720 BA Bilthoven, Netherlands
[4] Energy Research Centre of the Netherlands, P.O. Box 1, 1755 ZG Petten, Netherlands
[5] Centre for Isotope Research, Energy and Sustainability Research Institute Groningen, University of Groningen, Nijenborgh 4, 9747 AG Groningen, Netherlands

*Correspondence to*: Ingrid Super (ingrid.super@wur.nl)

**Abstract.** Monitoring urban-industrial emissions is often challenging, because observations are scarce and regional atmospheric transport models are too coarse to represent the high spatiotemporal variability in the resulting concentrations. In this paper we apply a new combination of a Eulerian model (WRF with chemistry) and a Gaussian plume model (OPS). The modelled mixing ratios are compared to observed $CO_2$ and CO mole fractions at four sites along a transect from an urban-industrial complex (Rotterdam, Netherlands) towards rural conditions for October–December 2014. Urban plumes are well-mixed at our semi-urban location, making this location suited for an integrated emission estimate over the whole study area. The signals at our urban measurement site (with average enhancements of 11 ppm $CO_2$ and 40 ppb CO over the baseline) are highly variable due to the presence of distinct source areas dominated by road traffic/residential heating emissions or industrial activities. This causes different emission signatures that are translated into a large variability in observed $\Delta CO{:}\Delta CO_2$ ratios, which can be used to identify dominant source types. We find that WRF-Chem is able to represent synoptic variability in $CO_2$ and CO (e.g. the median $CO_2$ mixing ratio is 9.7 ppm (observed) against 8.7 ppm (modelled)) , but it fails to reproduce the hourly variability of daytime urban plumes at the urban site ($R^2$ up to 0.05). For the urban site, a plume model should be added to the model framework to adequately represent plume transport especially from stack emissions. The explained variance in hourly, daytime $CO_2$ enhancements from point source emissions increases from 30 % with WRF-Chem to 52 % with WRF-Chem in combination with the most detailed OPS simulation. The simulated variability in $\Delta CO{:}\Delta CO_2$ ratios decreases drastically from 1.5 to 0.6 ppb ppm$^{-1}$ which agrees better with the observed standard deviation of 0.4 ppb ppm$^{-1}$. This is partly due to improved wind fields (increase in $R^2$ of 0.10), but also due to improved point source representation (increase in $R^2$ of 0.05) and dilution (increase in $R^2$ of 0.07). Based on our analysis we conclude that a plume model with detailed and accurate dispersion parameters is inevitable for top-down monitoring of greenhouse gas emissions in urban environments with large point source contributions within a ~10 km radius from the observation sites.





## 1 Introduction

Cities are major contributors to anthropogenic $CO_2$ and air pollutant emissions (Brioude et al., 2013; Turnbull et
al., 2015; Velasco et al., 2014). Both monitoring and modelling of urban/regional concentrations of $CO_2$ and co-
emitted air pollutants, such as CO and $NO_x$, has therefore received a lot of attention (Brioude et al., 2013; Font et
al., 2014; Huszar et al., 2016; Lac et al., 2013; Mays et al., 2009; McKain et al., 2012; Rayner et al., 2014;
Ribeiro et al., 2016; Silva et al., 2013; Tolk et al., 2009; Wunch et al., 2009; Zhang et al., 2015). Since current
emission inventories at small scales contain substantial uncertainties (Pouliot et al., 2012; Vogel et al., 2013),
data assimilation has been applied to urban environments in order to better quantify fossil fuel fluxes. However,
modelling urban atmospheric composition remains challenging as the urban environment is complex in both the
emission landscape and atmospheric transport. This means that to independently estimate urban emissions from
atmospheric observations, urban inversions require a detailed and accurate transport model that allows the
mismatch between model and observations to be attributed to errors in the emission inventory, rather than to
transport errors (Boon et al., 2016). Previous inversion studies relied heavily on a strict data selection to favour
well-mixed conditions with more reliable model output, which results in very small data sets and therefore
increased uncertainty on the estimated emissions (Bréon et al., 2015; Brioude et al., 2013). This could be
overcome by improving the model representation of urban transport, taking into account that the model
requirements are strongly dependent on the type of observation site used in the inversion. In this paper we aim to
construct a promising observation and modelling framework to quantify the $CO_2$ budget of an urban area by
addressing two important questions in the context of inverse modelling at the urban scale.

The first question is what type of measurement location (urban vs. rural) can best be used to monitor urban
fluxes. Generally, urban sites are most strongly exposed to nearby (<1 km) fluxes and therefore show a large
variability (Bréon et al., 2015; Lac et al., 2013). In contrast, rural sites show a much smaller response to urban
emissions due to the small range of wind directions at which the site is affected by the urban area. Moreover, the
dilution of urban plumes increases with distance (Calabrese, 1990; Finn et al., 2007) and the observed signal at
the rural site can be small. Another consideration is that near-ground measurements, as commonly found in
cities, are highly influenced by local sources (<100 m) that mask the urban signal. Boon et al. (2016) suggested
that, even if strict data selection is applied, the usefulness of such sites in inversions with high-resolution
Eulerian models (1−10 km) might be limited. Together, these papers suggest that a useful measurement location
should be just downwind of an urban area relative to the dominant wind direction at a distance that ensures
enough exposure to the urban plume and limits model errors due to large heterogeneity and local emissions. We
will examine a transect of measurement sites to see which site best matches this criterion.

The second question we address is what type of modelling framework is best capable of explaining urban
transport and the resulting mole fractions at the measurement sites. Since the measurement location determines
the level of spatiotemporal variation that can be observed in the concentrations, it also determines the
requirements imposed on the modelling framework. In atmospheric composition modelling both Eulerian and
Lagrangian (plume, puff or Gaussian) models are used, or a combination of both (Kim et al., 2014; Korsakissok
and Mallet, 2010a). Eulerian models use a grid that can be adapted to cover either small or large areas at
different resolutions and are therefore widely used. However, Eulerian models assume that trace gasses are
instantly mixed within individual grid boxes, which may enhance dispersion in the horizontal and vertical. The
resulting errors in transport and mixing are reflected in unrealistic concentrations (Karamchandani et al., 2011;





Tolk et al., 2009). The magnitude of the concentration error depends on the heterogeneity of the emissions and the grid resolution (Tolk et al., 2008). A plume model improves the description of horizontal and vertical mixing and can account for higher spatial heterogeneity of emissions and concentrations. The use of such models has proven useful for both inert and reactive species, and point and line sources at local/urban scales (Briant and Seigneur, 2013; Korsakissok and Mallet, 2010a, b; Rissman et al., 2013; Vinken et al., 2011). However, a plume model is usually only applied to local sources to reduce computational expenses. It therefore does not resolve the impact of remote emissions and synoptic transport. So, when assessing the carbon balance of a whole city or larger areas, a combination of both models might be needed.

Oney et al. (2015) examined an extensive $CO_2$, $CH_4$ and CO measurement network in combination with the FLEXPART-COSMO model. However, their framework focused on regional (~100–500 km), terrestrial fluxes. Several other studies focussed on urban scales (Boon et al., 2016; Bréon et al., 2015), but only few incorporated a Lagrangian model. For example, McKain et al. (2012) used the Lagrangian STILT model to optimize urban fluxes of $CO_2$, while Brioude et al. (2013) compared simulated FLEXPART $CO_2$, CO and $NO_x$ concentrations to small observational datasets from seven flights over Los Angeles. Here, we compare and combine simulations with two different models: the Eulerian WRF-Chem model and the segmented Gaussian plume model OPS. The Gaussian plume model is used here specifically to transport point source emissions. The model output is compared to continuous observations of $CO_2$ and CO at several measurement sites along an urban-to-rural transect. We included CO, because this species can act as a useful tracer for source attribution. We use the Rijnmond area (The Netherlands) including the city of Rotterdam as our case study, which is surrounded by scattered urban, agricultural, and rural areas. We chose this area because of the availability of a 1x1 km$^2$ emission inventory and its complex combination of residential, transport (including shipping), greenhouse and industrial activities. This makes Rijnmond an interesting test case, albeit not a simple one.

This paper starts with a description of the case study (Sect. 2.1), the modelling framework (Sect. 2.2–2.5), and a summary of data selection criteria and methods (Sect. 2.6). Subsequently, we examine the ability of our measurement sites to detect urban signals, and demonstrate the added value of both urban and semi-urban sites (Sect. 3.1). Section 3.2 examines the ability of WRF-Chem to represent the urban signals at the measurement sites. Finally, we discuss the advances made by implementing the Gaussian OPS plume model (Sect. 3.3) and we examine the relative importance of improved meteorological conditions and source representation in Sect. 3.4. Our results lead to recommendations for future monitoring and modelling of urban atmospheric composition in Sect. 4.

## 2 Methods

### 2.1 Study area and measurements

We take the Rijnmond area (Fig. 1) in the Netherlands for our case study in which Rotterdam is the major urban area (625.000 inhabitants). The area is situated in flat terrain near the west coast of The Netherlands and includes a large harbour and industrial area. The bottom-up estimated emissions in this area are about 35 Mt $CO_2$ and 48 kt CO in 2012 (Netherlands PRTR, 2014). In the port area, over three times more $CO_2$ is emitted than in the city of Rotterdam. In contrast, more than 60 % of all CO is emitted in the city of Rotterdam. The reason for this difference is that emissions within the city are dominated by road traffic, which emits relatively much CO





(CO:CO$_2$ emission ratio of almost 17 ppb ppm$^{-1}$). The principal source of CO$_2$, namely energy production and industrial processes, is mainly found in the port area and barely emits any CO (CO:CO$_2$ emission ratio of less than 1 ppb ppm$^{-1}$). The CO$_2$ emissions are therefore dominated by point sources (~80 %).

We have installed two measurement sites to monitor CO$_2$ and CO mixing ratios 15 km south (Westmaas, 51.79° N, 4.45° E) and 7 km northwest (Zweth, 51.96° N, 4.39° E) of the city centre with an inlet at 10 m a.g.l. We consider Zweth to be an urban site which is highly affected by urban emissions. Westmaas functions as a background site close to – but not within - the city and it is usually located upwind of the major source areas. Therefore, Westmaas provides information on the air mass entering the Rijnmond area and we only use this site to validate the large-scale patterns in WRF-Chem. These measurements have been described in more detail by Super et al. (2017). At Rotterdam-The Hague airport (Fig. 1) meteorological observations are made, which we also use for transport model validation purposes.

We include two additional, more remote, sites in our framework. The Cabauw site (51.97° N, 4.93° E) is situated 32 km east of the centre of Rotterdam and is considered a semi-urban site (Van der Laan et al., 2016; Vermeulen et al., 2011). This means the sampled air masses are influenced by urban emissions, but less often than a truly urban location. CO$_2$ is measured at several heights (20, 60, 120 and 200 m a.g.l.) along a 200 m tall tower by the Energy research Centre of the Netherlands (ECN). CO is measured at ground level (2.5–4 m a.g.l.) by the National Institute for Public Health and the Environment  (RIVM). Another observation site is located at Lutjewad (53.40° N, 6.35° E), close to the coast in the north of the Netherlands. At this rural site, CO and CO$_2$ mixing ratios are observed at 60 m a.g.l. (Van der Laan et al., 2009a; Van der Laan et al., 2016). These four stations together describe a transect from the city towards rural areas.

For the Cabauw CO$_2$ measurements we selected the 60 m level. On average the CO$_2$ mixing ratios are similar at all levels during well-mixed daytime conditions (Vermeulen et al., 2011), but a large gradient is observed for stable conditions when the 20 m level is highly affected by surface fluxes surrounding the tower. Similarly, Turnbull et al. (2015) suggested that measurements closer to the surface are more sensitive to local fluxes and therefore a higher level than 20 m is more suitable to obtain information on more remote fluxes. We choose the 60 m level observations to be able to compare easily to the Lutjewad site. However, a higher level could have been used without affecting our conclusions.

### 2.2 Eulerian model

The Eulerian model used in this study is WRF-Chem V3.2.1 (Skamarock et al., 2008). For its initial and boundary conditions we use meteorological fields from the National Centers for Environmental Prediction (NCEP) Final (FNL) Operational Global Analysis (National Centers for Environmental Prediction/National Weather Service/NOAA/U.S. Department of Commerce, 2000) at 1x1° horizontal resolution and a temporal resolution of 6 hours. We define four 2-way nested domains (Fig. 2) which have a horizontal resolution of 48x48, 12x12, 4x4 and 1x1 km respectively, and a vertical resolution of 29 eta levels with the lowest model layer 40 m deep and a total of 8 levels in the lowest 1 km. The outer domain is situated over Europe. Domains 2–4 zoom in on the Rijnmond area in the southwest of the Netherlands. Based on previous studies over the Netherlands (Bozhinova et al., 2014; Daniels et al., 2016; Steeneveld et al., 2014), we have used the Yonsei University (YSU) boundary layer scheme (Hong et al., 2006), the Dudhia scheme for shortwave radiation (Dudhia, 1989), the Rapid Radiation Transfer Model (RRTM) as longwave radiation scheme (Mlawer et al.,





1997), and the Unified Noah Land-Surface Model as the surface physics scheme (Ek et al., 2003). We also used the single-layer urban canopy model (UCM) to account for changes in roughness length and heat fluxes in the urban environment (Chen et al., 2011), although the impact of the UCM model on simulated mixing ratios is very small in our domain.

The $CO_2$ initial and boundary conditions are taken from the 3D mole fractions from CarbonTracker Europe (Peters et al., 2010). The CarbonTracker 3D fields have a horizontal resolution of 1x1° and 34 vertical levels. Therefore, they are both horizontally and vertically interpolated onto the WRF-Chem grid. The CO initial and boundary conditions are calculated with IFS-MOZART (Flemming et al., 2009) and obtained from the Monitoring Atmospheric Composition and Climate (MACC) project. The boundary conditions are updated every 6 hours (only for the outer domain).

We have implemented a $CO_2$ budget based on the methodology used by Bozhinova et al. (2014), described in Eq. (1).

$$X_{CO2,obs} = X_{CO2,lsbg} + X_{CO2,ff} + X_{CO2,bf} + X_{CO2,p} + X_{CO2,r} \qquad (1)$$

where the indices express the origin of $CO_2$: obs – total observed concentration at a particular location, lsbg – large-scale background mole fraction, ff – fossil fuels, bf – biofuels, p – photosynthetic uptake, r – ecosystem respiration. Similar to the original study of Bozhinova et al. (2014), we omitted the stratosphere-troposphere exchange and ocean fluxes and assume they are accounted for in the large-scale background. With Eq. (1) we thus only consider regional contributions to the carbon budget in addition to the large-scale background. In the model, any change in the large-scale background $CO_2$ mole fraction ($X_{CO2,lsbg}$) is only caused by advection and exchange at the domain boundaries.

In addition, we added the CO budget to WRF-Chem following Eq. (2). The main sources of CO are fossil fuel combustion and oxidation of hydrocarbons (US EPA, 1991). Several scholars have argued that the hydrocarbon oxidation term is important for the large-scale background CO concentration (Gerbig et al., 2003; Griffin et al., 2007; Hudman et al., 2008), contributing a significant percentage to the total CO burden. Yet, these studies were all based on summer time measurements and under conditions favourable for photochemistry. Photochemical oxidation is likely to be less important in the winter months considered here. Moreover, Gerbig et al. (2003) found the CO fraction from local anthropogenic emissions to dominate at measurement sites. We assume this is also valid in the urban-industrial environment of our case study.

The main sink of CO is the reaction with the hydroxyl radical (chemical loss term $L$), which we account for with a simple first order loss term. We assume steady-state, i.e. the OH concentration is taken as a constant ($10^6$ molecules cm$^{-3}$). This results in a lifetime for CO of about 2 months at mid-latitudes (Jacob, 1999) during the winter months used in our study:

$$X_{CO,obs} = X_{CO,lsbg} + X_{CO,ff} + X_{CO,L} \qquad (2)$$

Biofuel fluxes for CO are not known for the study area. The different contributions in Eq. (1) and Eq. (2) are separated as different additive tracers (i.e. labelled) in the WRF-Chem simulations.

**2.3 Gaussian plume model**

The plume dispersion model OPS (Operational Priority Substances) is a segmented Gaussian plume model that calculates the transport, dispersion, chemical conversion and deposition of pollutants (Sauter et al., 2016; Van Jaarsveld, 2004). It is used to calculate large-scale, yearly averaged concentration and deposition maps for the



Netherlands at 1x1 km$^2$ resolution. It was initially developed to model dispersion of pollutants like particulate

matter and ammonia, but has also been used to study the dispersion of pathogens (Van Leuken et al., 2015).

In this paper we use the so-called "short-term" version of this model (version 10.3.5), which contains mostly the same parameterisations as the "long-term" model described by Sauter et al. (2016). The short-term model provides hourly concentrations at receptors that can be individual sites, or across a gridded domain. The model keeps track of a trajectory, for which plumes consist of so-called segments, taking into account time-varying

transport over longer distances (e.g. changes in wind direction and dispersion). If for a time step a specific plume affects the receptor, a Gaussian plume formulation is used to calculate the concentration caused by that source based on the true travel distance along the trajectory.

The OPS model uses primary meteorological variables which are measured by the Royal Dutch Meteorological Institute, and calculates secondary variables such as boundary layer height and friction velocity, but also the

turning of the wind with height and a vertical wind profile. Primary meteorological variables are spatially interpolated over the Netherlands to 10x10 km$^2$ using 19 observation sites with a weighing factor depending on the distance to the grid point. The variables are subsequently averaged over a pre-defined area (for more information see Sauter et al. (2016)). The use of observed meteorology in OPS versus model-calculated meteorology in WRF-Chem could result in an unfair comparison of the models, and we therefore replaced the

primary parameters (temperature, humidity, wind speed, and wind direction) and the boundary layer height with those calculated by WRF-Chem. The secondary (dispersion) parameters are automatically also updated, since they are calculated from the primary parameters. Note that the meteorological conditions in OPS remain constant during each simulated hour and over a large region.

Although potentially the OPS model can be used for both area and point source emissions, we believe that

point sources will benefit most from a more detailed description of dispersion as they are affected most by the instant dilution in a Eulerian model. When using OPS, we assume wet deposition plays no role due to the relative insolubility of $CO_2$, while dry deposition of $CO_2$, i.e. photosynthetic uptake, is accounted for by WRF-Chem (Eq. (1)). We do not simulate CO with the OPS model. The point source contribution to the total CO concentrations is very small and therefore the impact of OPS is limited.

**2.4 Emissions**

The fossil fuel and biofuel emissions for domains 1–3 in the WRF-Chem simulation are taken from the TNO-MACC III inventory for 2011 (Kuenen et al., 2014) and have a horizontal resolution of 0.125x0.0625°. Fossil fuel emissions for domain 4 in WRF-Chem are collected from the Dutch Emission Registration (Netherlands PRTR, 2014) and compiled by TNO (Netherlands Organization for Applied Scientific Research) to a 1x1 km$^2$

emission map for the year 2012. In the OPS simulations we only include the point source emissions from domain 4 in WRF-Chem (hereafter referred to simply as the Rijnmond area).

The emissions are divided over ten SNAP emission categories, summarised in Table 1, which may include both area and point sources. We apply a temporal profile to the emissions by assigning hourly, daily and monthly fractions to the emissions per emission category (Denier van der Gon et al., 2011). In WRF-Chem, area source

emissions are added to the lowest surface model level every hour. Point source emissions (only SNAP 1, 3, 4, 8 and 9) are given a simplified, fixed vertical distribution based on previous research with plume rise calculations (Bieser et al., 2011). These emissions are emitted at the heights shown in Table 1. OPS allows for more detailed



point source characteristics and accounts for stack height and plume rise (based on heat content) per individual point source.

235 The biogenic (non-biofuel) $CO_2$ fluxes in WRF-Chem are generated as described by Bozhinova et al. (2014). The SiBCASA model (Schaefer et al., 2008) calculates monthly averaged 1x1° photosynthetic uptake ($A_n$) and ecosystem respiration ($R$) for nine different land use types. Combining the high-resolution land-use map of WRF-Chem with the SiBCASA fluxes gives us biogenic fluxes on the resolution of the WRF-Chem grid. The temporal resolution is enhanced by scaling the $A_n$ and $R$ at each WRF-Chem time step with modelled shortwave

240 solar radiation ($SW_{in}$ in W m$^{-2}$) and 2m temperature ($T_{2m}$ in K):

$$A_n = A_{n,f} \cdot SW_{in} \tag{3}$$

$$R = R_f \cdot 1.5^{(T_{2m}-273.15)/10} \tag{4}$$

where $A_{n,f}$ is the monthly average photosynthetic flux divided by the monthly total incoming shortwave radiation (mole $CO_2$ km$^{-2}$ h$^{-1}$ (W m$^{-2}$)$^{-1}$), and $R_f$ the monthly average respiration flux (mole $CO_2$ km$^{-2}$ h$^{-1}$) divided by the

245 monthly total of the empirical function $1.5^{(T_{2m}-273.15)/10}$ (unitless). This procedure was first described in Olsen and Randerson (2004). It neglects the impact of water stress, temperature and $CO_2$ concentration on the photosynthetic uptake. Given that we consider only winter months in which photosynthesis is limited, we assume the error resulting from this simplification to be small.

**2.5 Overview of simulations**

250 We simulated a period of 3 months, October–December 2014. We choose this period because of the high data coverage at all measurement sites and to limit the impact of biogenic fluxes and hydrocarbon oxidation. We considered four simulations for $CO_2$, using two different model systems as described in Table 2. All simulations include the WRF-Chem contributions of $X_{CO2,lsbg}$, $X_{CO2,p}$, $X_{CO2,r}$, $X_{CO2,bf}$, and $X_{CO2,ff}$ from area sources. Also, the first three simulations make use of meteorological conditions as simulated by WRF-Chem. Therefore, the

255 simulations only differ in the representation of point source emissions in the Rijnmond area. To identify the importance of a correct representation of meteorological conditions we do an additional OPS simulations with interpolated meteorological observations (see Sect. 2.3). The simulations are designed to gradually increase the complexity of the point source representation towards more realistic point source contributions:

- In simulation 1 (WRF-Chem) the point sources are represented as area sources in WRF-Chem;
260 - In simulation 2 (WRF+OPS-area) the point sources are treated as area sources in OPS;
- In simulation 3 (WRF+OPS-point) the point sources are represented as true point sources with detailed source characteristics in OPS;
- In simulation 4 (WRF+OPS-point-obsmet) the point sources are represented as true point sources with detailed source characteristics in OPS and the meteorology in OPS is replaced by interpolated
265 observations and OPS calculated boundary layer height.

In the WRF-Chem run we labelled the point source emissions from the Rijnmond area separately, so we can replace them by the OPS counterparts. The OPS model simulates concentrations directly at the measurement sites, whereas from WRF-Chem we extract the grid box average mixing ratio of the boxes in which the measurement sites are located.

270 **2.6 Baseline determination and data selection criteria**



In this study we are especially interested in the contribution of urban emissions and the ability of the models to represent the transport of those emissions to the observation sites. However, the observed $CO_2$ and CO mixing ratios are also affected by background signals and other fluxes. Therefore, in order to purely compare the transport of urban emissions, we need to separate the fossil fuel contribution from all other contributions. In the

models we can separate the fossil fuel contribution $X_{CO2,ff}$ coming from the Rijnmond area (hereafter referred to as "urban plume") from all other contributions (i.e. $X_{CO2,ff}$ from outside the Rijnmond area, $X_{CO2,lsbg}$, $X_{CO2,p}$, $X_{CO2,r}$, and $X_{CO2,bf}$, hereafter referred to as "baseline") by using labelled tracers. To quantify the urban plume contribution to the total observed mixing ratio, we also need to subtract a baseline.

Previous studies have suggested various methods to calculate the baseline from observations, for example

using a remote/upwind measurement site or statistical methods (e.g. (Djuricin et al., 2010; Lopez et al., 2013; Turnbull et al., 2015; Van der Laan et al., 2010). An in-model comparison with WRF-Chem shows that Westmaas is a suitable background site for Zweth (Super et al., 2017), but Westmaas gives a biased baseline estimate for the more remote sites (Cabauw and Lutjewad) because of the interference of other sources and sinks along the transect from Rijnmond to the measurement site. Another suggested method is to subtract a smoothed

representation of the original time series (Press et al., 1992; Super et al., 2017; Thoning and Tans, 1989) which filters out variations below a certain cut-off time scale. For seasonal cycle smoothing for example, a typical cut-off value is 80 days. In our study however, the baseline needs to filter out synoptic variations across the domain and we therefore chose a cut-off time of 5 days. We tested this baseline definition by applying it to the WRF-Chem time series and comparing the resulting concentrations to the true WRF-Chem baseline based on the

labelled tracers. We found satisfactory agreement ($R^2$ is between 0.65 and 0.81 for both species at all 3 locations).

To prevent any differences between model and observations resulting from the baseline selection, we choose to apply this subtraction of a smooth cycle method with a 5-day cut-off to both observations and our model time series at all measurement sites (see Fig. 3 for an example). The concentrations above the baseline are considered

to be the urban plume concentrations and are denoted $\Delta CO_2$ and $\Delta CO$. Note that data points can also be below the baseline if clean air is advected and only a small fossil fuel contribution is calculated. We discard these data points, because we cannot accurately estimate the fossil fuel concentrations in those urban plumes.

In all the analyses, we applied a wind sector selection to ensure that the observations are affected by emissions in the Rijnmond area rather than from other urban areas nearby. For Zweth we selected wind directions of 90–

220 degrees, for Cabauw 230–270 degrees, and for Lutjewad 210–230 degrees. For Zweth we can also separate between signals from the residential area (90–150 degrees, Zweth-city) and industrial area (160–220 degrees, Zweth- port). Wind direction observations at Rotterdam airport are used for this purpose. Additionally, a daytime selection criterion (8:00–17:00 LT) is applied to favour well-mixed conditions.

## 3 Results

### 3.1 Comparison of measurement sites

The urban-to-rural transect of observation sites provides an opportunity to evaluate the ability of different types of sites to detect urban plumes. We find that a semi-urban site can provide a constraint on the total emissions in the Rijnmond area, whereas an urban site is able to separate between different source areas. This is illustrated in




Fig. 4 (left panel), where we display the probability density functions of the urban plume $CO:CO_2$ concentration
ratio (i.e. $\Delta CO:\Delta CO_2$) at the three sites. A probability density function illustrates the likelihood that an observed
urban plume concentration ratio takes a certain value. The narrower the distribution, the less variable the ratios
are and the more likely a ratio is to take the mean value (largest probability). Figure 4 also displays the mean
bottom-up derived emission ratio of the Rijnmond area (vertical solid line, 2.5 ppb ppm$^{-1}$) and its range, which is
taken from the emission inventory taking into account the temporal profiles of the separate emission categories.

315        We see that the $\Delta CO:\Delta CO_2$ distribution at Cabauw is relatively narrow. Also, the mean $\Delta CO:\Delta CO_2$ at Cabauw
(2.2 ppb ppm$^{-1}$) is very close to the bottom-up Rijnmond emission ratio. This indicates that Cabauw observes an
integrated, well-mixed signal from the Rijnmond area and therefore contains information on the entire urban
area. Interestingly, Lutjewad shows a much wider distribution with a mean of 3.9 ppb ppm$^{-1}$. The urban plume
from Rijnmond is mixed with signals from other industrial and urban areas (such as Amsterdam) before it
reaches Lutjewad, causing more variability. This suggests that a site too far away from the urban sources is
unable to uniquely identify the urban plume coming from a specific region. Also, the wind direction is
heterogeneous between Rijnmond and Lutjewad. So, despite that the wind in Rijnmond is blowing towards
Lutjewad according to our wind sector selection, the urban plume might never reach the site if the wind direction
is changing during transport. This makes it difficult to filter out the Rijnmond urban plume and Lutjewad will be
disregarded for the remainder of this study. The Zweth site has an even wider distribution than Lutjewad and a
mean ratio of 4.5 ppb ppm$^{-1}$. This site is affected by different source areas with distinct emission ratios
depending on the wind direction, resulting in a large variability in observed concentration ratios. This variability
contains a lot of information about the Rotterdam emissions and their spatiotemporal variations. Therefore, we
examine the Zweth distribution in more detail by selecting wind sectors that sample different source areas with
distinct emission characteristics (Fig. 4, right panel). Zweth-city is illustrative for the signal from the urban
residential area dominated by road traffic and the Zweth-port signal contains mostly industrial and power plant
emissions.

    We find a large difference in bottom-up emission ratios for the residential (6.6 ppb ppm$^{-1}$, vertical dash-dotted
line) and port area (1.2 ppb ppm$^{-1}$, vertical dashed line), which is not fully reproduced by the observed
$\Delta CO:\Delta CO_2$ ratios. Whereas the observed $\Delta CO:\Delta CO_2$ ratio for Zweth-city (5.0 ppb ppm$^{-1}$) is in reasonable
agreement with the emission ratio, Zweth-port has a mean observed ratio that is much higher than expected (4.1
ppb ppm$^{-1}$). This discrepancy is related to the presence of high stack emissions in this area, which make up
almost 75 % of the total Rijnmond $CO_2$ emissions. The stack emissions from industrial processes and energy
production have a small emission ratio of ~1 ppb ppm$^{-1}$ and dominate the total emission ratio. However, stack
emissions have small plume dimensions that can easily be missed at the Zweth site and not be visible in the
observations, especially for stacks in the vicinity of Zweth. Therefore, the observed concentration ratio can turn
out much higher than what is expected based on the emission inventory including stack emissions. Indeed, the
emission ratio of the Zweth-port area without point sources would be 3.9 ppb ppm$^{-1}$, which is very close to the
observed 4.1 ppb ppm$^{-1}$. This finding indicates that stack emissions only occasionally affect the Zweth
observations and it is very important to represent those events well with a model in order to constrain this large
fraction of $CO_2$ emissions. The impact of stack emissions on the Zweth observations will be discussed in more
detail in Sect. 3.3.

**3.2 WRF-Chem urban plume transport**



We have now seen that the observations at Zweth and Cabauw contain valuable information about the emissions
in the Rijnmond area. In order to use that information to estimate the emissions, we explore the ability of WRF-
Chem to represent observed time series, and especially their urban plume components.

First, we analyse the model performance on a day-to-day basis by looking at daytime averages and find that
WRF-Chem is able to resolve day-to-day variations reasonably well. Table 3 shows that, respectively, 65 % and
53 % of the variability in the $CO_2$ and CO mixing ratios is captured at the Westmaas background site. Although
the explained variances are slightly smaller at the urban (Zweth) and semi-urban (Cabauw) site, the performance
at Cabauw for $CO_2$ is comparable to previous modelling studies (Bozhinova et al., 2014; Tolk et al., 2009). Yet,
the RMSE is relatively large for CO and $CO_2$ at all sites. Since Westmaas is nearly unaffected by urban
emissions, the cause of the large RMSE is related to larger scale transport. Looking at meteorological variables,
there is a good agreement for temperature, humidity and wind speed. However, the model has difficulties
simulating the correct wind direction, which is especially expressed in the large RMSE. The largest error is
found in the second half of November, causing a large model-data discrepancy (also visible in Fig. 3). Table 3
also shows that the RMSE in the mixing ratios further increases for sites that are more influenced by the urban
area. This finding indicates that WRF-Chem has difficulties representing the full variability caused by urban-
industrial emissions.

Second, looking closer at the urban plumes we find that WRF-Chem represents the typical characteristics of
urban plumes reasonably well, but it simulates the peaks at the wrong time at the wrong location compared to the
measurements (Table 4). We tried to isolate the impact of errors in urban transport by looking statistically (i.e.,
there is no co-sampling in time for this comparison) at the urban plume concentrations ($\Delta CO_2$ and $\Delta CO$) at
Zweth and Cabauw. We disregard data points associated with wind speeds of less than 3 m s$^{-1}$ to favour well-
mixed conditions that are easier to interpret. Table 4 shows that, on average, there is a good agreement between
WRF-Chem and the observations. The median values are generally somewhat lower in WRF-Chem, indicating
there are more small values and less high peak values in the model. Also the 80$^{th}$ percentile shows reasonable
agreement. Because the frequency distribution of the wind direction is similar between the observations and
WRF-Chem, we expect no bias is introduced by the wind direction error. However, if we co-sample WRF-Chem
and the observations over time when the observations match our criteria (note that this creates a different subset
of samples) we find a very small explained variance ($R^2$) for both species at both sites based on hourly data. An
inversion using these hourly data would thus be subject to a large model-data mismatch that increases the
uncertainty in the optimized fluxes. Therefore, we next look more specifically at the data points responsible for
the highest mismatch in observed and simulated $\Delta CO_2$.

We find that the largest differences between WRF-Chem and the observations at Zweth when co-sampling
urban plumes results from errors in simulated wind direction, as well as from an inability of WRF-Chem to
simulate the impact of point source emissions. This is illustrated in Fig. 5, where we binned the absolute errors in
hourly $\Delta CO_2$ into four magnitude classes of 10 ppm each and correlate them with the error in simulated wind
direction (as binned into three classes of 20 degrees, scatter plots) and with the observed $\Delta CO:\Delta CO_2$ ratio
(whisker plots). We find that the smallest $\Delta CO_2$ model error class (0–10 ppm) is dominated by the smallest wind
direction error (0–20 degrees, 72 %), while in the largest $\Delta CO_2$ model error class (30–40 ppm) almost 50 % of
the data points have a wind direction error of more than 20 degrees. With such large wind direction errors, the
trajectory of urban plumes is misrepresented and the modelled mixing ratios are affected by the wrong source





area, or plumes may even entirely miss the sites in the model. In addition, we find that in the largest $\Delta CO_2$ model
error class (30–40 ppm) the observed $\Delta CO:\Delta CO_2$ is lower (2.5 ppb ppm$^{-1}$) and less variable than in the other
classes, suggesting a larger influence of industrial (stack) emissions. Although the number of data points in the
largest $\Delta CO_2$ model error class is small (N=14), these tendencies give a good indication of what might cause
these errors. At Cabauw, the impact of stack emissions is not visible, because the point source emissions are
already well-mixed when the air mass arrives at Cabauw. Hence, we will next examine the added value of the
OPS plume model at Zweth to better represent the dispersion of $CO_2$ emitted from stacks and the impact of wind
direction in OPS.

### 3.3 WRF-Chem and OPS point source representation

When we focus exclusively on point source emissions, we find that all simulations that include the OPS plume
model are in better agreement with the observations than the WRF-Chem simulation (based on the $R^2$ and
regression slope). This is illustrated in Table 5, where we compare co-sampled simulated and observed events
with a high point source contribution. These events are selected based on a low observed $\Delta CO:\Delta CO_2$ ratio (the
threshold is 1.5 ppb ppm$^{-1}$, events illustrated as grey bars in Fig 4). In the models, these events are highly
correlated with a high point source contribution (of at least 90 %) in the simulated $\Delta CO_2$ mixing ratio ($r$ is -0.76
(WRF+OPS-point-obsmet) and -0.61 (WRF-Chem)). For WRF-Chem the explained variance in the co-sampled
observations is limited ($R^2$=0.30) and the regression slope of $\Delta CO_2$ is significantly lower than one (i.e. the 1:1
line of modelled vs. observed $\Delta CO_2$). Both the mean $\Delta CO:\Delta CO_2$ and the standard deviation are larger than the
observed mean and standard deviation. This suggests that the lack of agreement is partly caused by an error in
the WRF-Chem wind direction, causing the model to sample air from a wrong source area.

In contrast to WRF-Chem, WRF+OPS-point-obsmet shows a larger explained variance ($R^2$=0.52), a regression
slope that is nearly one, and a $\Delta CO:\Delta CO_2$ ratio that agrees with observations both in mean and in standard
deviation. Since only about 10 % of the Zweth-port observations are affected by stack emissions due to the small
dimension of the plumes (N=42), a better representation of atmospheric conditions has a large impact. An
advantage of the OPS model is the ability to estimate the model uncertainty by providing a plume cross-section.
Receptor points can be positioned anywhere and by adding several receptor points around the true measurement
location we can account for transport errors (e.g. in the wind direction). If we allow for a maximum wind
direction error of 5 degrees, this has no significant impact on the $R^2$ or slope, suggesting that the results from the
WRF+OPS-point-obsmet simulation are robust against small random errors in wind direction. However,
systematic errors in the wind direction or the treatment of point source emissions such as present in WRF-Chem
will have an impact on its performance, as we will explore next.

### 3.3.1 Dispersion

When comparing WRF-Chem and WRF+OPS-area we find that the OPS model reduces the dispersion of point
source emissions, which causes emissions from high stacks to barely reach ground level. Vertical profiles of
$\Delta CO_2$ near an energy production stack for both model simulations are shown in Fig. 6. Energy production
sources often have the highest stacks and the lowest $\Delta CO:\Delta CO_2$ ratios. Near an energy production stack the
vertical dimension of the plume in WRF+OPS-area is smaller than in WRF-Chem. The plume remains more
concentrated in WRF+OPS-area, leading on average to lower mixing ratios at ground level (left panel) and to



higher maximum values at around 200 m (right panel). This effect is also clearly visible at Zweth (not shown) and results in a higher mean $\Delta CO:\Delta CO_2$ ratio in Table 5 for WRF+OPS-area (i.e. less influence of the low-ratio stack emissions) and a higher explained variance (37 %).

**3.3.2 Point source representation**

From a comparison of WRF-Chem, WRF+OPS-point and WRF+OPS-point-obsmet it follows that having a plume model with full point source characteristics can improve the agreement with the observed mixing ratios, even if the meteorological conditions are biased. Implementing detailed source characteristics (WRF+OPS-point) not only increases the explained variance to 42 %, it also increases the $\Delta CO:\Delta CO_2$ standard deviation. This is the result of larger spatial (both horizontal and vertical) variability in the emission landscape. These effects are also visible in Fig. 7, which shows a time series of six days of observations and model output. When differences between the simulations are small, this indicates the absence of point source signals. On October 23 (event A) an improvement is made by using observed meteorological conditions due to the large wind direction error, while the difference between WRF-Chem and WRF+OPS-point is small. However, on other occasions the use of the OPS model, irrespective of the meteorology used, already improves the simulated urban plume mixing ratio. For example, on October 24 (event B) both OPS runs reduce the urban plume mixing ratios and are in better agreement with the observations. On October 26 (event C) the opposite is happening. Whereas WRF-Chem is only above the background for four hours, the observations show a longer and more severe pollution event, despite a relatively small wind direction error. Although an additional improvement can be made using the observed wind fields, using WRF+OPS-point already improves the length and strength of the pollution event. Note that, although WRF-Chem sometimes performs better than the simulations including OPS, the overall statistics suggest that it is recommended to use WRF+OPS-point-obsmet.

**4 Discussion**

In this study we focused on two major questions in urban greenhouse gas modelling studies: what type of measurement locations can provide the best information on urban fluxes of $CO_2$ and CO, and what type of modelling framework can best represent urban plume mixing ratios at these measurement sites. In a previous study, Lauvaux et al. (2016) have used nine observation towers to estimate $CO_2$ fluxes from Indianapolis. They have argued that the optimum number of towers is dependent on the spatial heterogeneity of the emissions within the city. They also state that it is impossible to attribute changes in the total $CO_2$ concentration to specific source sectors when only $CO_2$ observations are available. Based on our current findings, we believe that with the use of other co-emitted species, like CO, information can be gained about source sector contributions, as was also shown by Turnbull et al. (2015). Additionally, Brioude et al. (2013) have shown that with only a few flights a reasonably robust flux estimate can be made for CO and $NO_y$. These studies thus show that with additional species and strategically placed measurements the need for a large observation network can be reduced. However, an important pre-condition is that atmospheric transport is correctly represented. Lauvaux et al. (2016) discussed that the atmospheric transport in high-resolution Eulerian models might suffer from errors due to assumptions about turbulence and other fine-scale processes, which causes urban plumes to violate the well-mixed assumptions of the model. This is especially relevant for emission sources with dimensions that are



significantly smaller than the model resolution, i.e. point sources. Indeed, in this study we find that a plume
model is required to correctly represent the transport of emissions from large point sources.

### 4.1 Comparison of observation sites

Since the demand put on the model performance depends on the measurement sites used in the inversion, we first
examined the use of the measurement sites to detect urban plumes. At the rural site (Lutjewad), the urban plume
has become mixed with other signals and the urban plume is difficult to distinguish. This site (at ~200 km from
the Rijnmond area) is therefore too far removed to constrain specifically the Rijnmond emissions, although it
was shown to constrain emissions from the larger urban conglomerate of the Randstad quite well (Van der Laan
et al., 2009b; Van der Laan et al., 2010). The semi-urban site (Cabauw) detects urban plumes from Rijnmond
which have already become well-mixed during transport. Moreover, the mean concentration ratio matches well
with the emission ratio for the Rijnmond area. We therefore argue that the Cabauw site could constrain the
overall emissions of the Rijnmond area due to its integrating power without the need for a multi-model approach.
In contrast, the urban location (Zweth) is highly exposed to the urban fluxes and is able to detect spatial
variations in emissions inside the urban area. We find distinct concentration ratios for different source areas that
can provide valuable information about dominant source types and areas. These findings are similar to a previous
study concluding that a network of in-city sites provides good constraints due to their high exposure and ability
to separate between different parts of the source area (Kort et al., 2013). However, the difference between the
emission ratio and observed concentration ratio for the Zweth-port area indicates that stack emissions might
frequently be missed at the Zweth measurement site due to the limited plume dimensions. Therefore, a correct
representation of the transport becomes increasingly important. Thus, we conclude that the Cabauw and Zweth
site have their own particular (dis)advantages and a combination of an urban and semi-urban site could be most
beneficial to constrain urban fluxes in detail.

### 4.2 Model skill

Next, we evaluated the skill of the Eulerian WRF-Chem set-up. The ability of our WRF-Chem framework to
represent daytime average mixing ratios is comparable with other model frameworks in the urban environment
(Bozhinova et al., 2014; Bréon et al., 2015; Lac et al., 2013; Tolk et al., 2009). However, WRF-Chem has a large
wind direction bias that makes it difficult to compare modelled and observed mole fractions. The monthly
average WRF-Chem wind direction shows an absolute bias of 1 (October), 51 (November) and 10 (December)
degrees compared to the observed wind direction at Rotterdam airport. The error in November is large compared
to previous findings (Jiménez et al., 2016) and this results in a large model-observation mismatch in tracer
mixing ratios (Fig. 3). Also at the Cabauw site, which is less influenced by build-up areas, the model-data
agreement for the 10 m wind direction in November is limited. Previous research has also shown an uncertainty
of 30–40 % in the tracer mixing ratio due to the uncertainty in meteorological conditions (Angevine et al., 2014;
Srinivas et al., 2016). Additionally, Angevine et al. (2014) have shown that using an ensemble mean of model
simulations with different meteorology does not necessarily lead to a better representation of plume transport and
dispersion in a Lagrangian model for area sources. We therefore speculate that assimilating observed wind fields
in WRF-Chem, as was done by Lauvaux et al. (2013), could be more beneficial to improve the modelled wind
fields and as such improve the plume transport.





Some studies argued that the main limitations of a Eulerian model are the enhanced dispersion due to instant mixing of species throughout the grid box and, related to that, the absence of a good point source representation (Karamchandani et al., 2011; Tolk et al., 2009). Our results show evidence for both limitations in the WRF-Chem set-up. First, WRF-Chem underestimates the median urban plume mixing ratios of both $CO_2$ and CO which should mainly be attributed to errors in transport and mixing. Whereas CO mixing ratios at the Zweth site are dominated by area sources, $CO_2$ mixing ratios are also highly affected by point source emissions. Therefore, their consistent underestimation cannot be caused solely by errors in point source emissions. Second, looking more specifically at the point source contribution, WRF-Chem can only explain 30 % of the variance and the spread in the $\Delta CO:\Delta CO_2$ ratio is too large compared to the observations. Thus the resolution appears to be too low to fully represent the transport of the urban plumes from point sources, similar to previous findings related to power plant plumes (Lindenmaier et al., 2014) and megacities (Boon et al., 2016).

In order to overcome the limitations of WRF-Chem related to point source representation and wind field errors, we evaluated the use of the OPS plume model with full point source characteristics and observed meteorological conditions. Several previous plume modelling studies with different species showed improvements compared to the gridded approach (Briant and Seigneur, 2013; Ganshin et al., 2012; Karamchandani et al., 2006; Karamchandani et al., 2012; Korsakissok and Mallet, 2010a; Rissman et al., 2013). In this study we find a significant improvement with WRF+OPS-point-obsmet at Zweth, both in the explained variance and the $\Delta CO:\Delta CO_2$ ratio. Also the observed-vs-simulated regression slope of the point source $\Delta CO_2$ mixing ratio becomes nearly one. In this analysis the number of selected data points is relatively small, because stack emissions can easily be missed by an observation site due to the small plume dimensions. Therefore, only a few events can be used to constrain point source emissions and a good representation of the plume transport is essential. Although there are only ~100 individual point sources in the Rijnmond area, they make up about 75 % of the total $CO_2$ emissions. Thus we argue that in an urban-industrial area with a significant point source contribution the use of a plume model is critical to get a reliable emission estimate. Further improvements can possibly be made by representing traffic emissions as line source emissions in a plume model (Briant and Seigneur, 2013) rather than considering them as gridded area sources in the Eulerian model.

Although part of the OPS-driven improvement can be attributed to the use of observed meteorological conditions, we have shown with the WRF+OPS-point simulation that there is also an improvement in point source representation. We found that a higher spatial variability in the emissions causes more variability in the concentration ratios. Representing point sources as area sources, as was done in WRF-Chem, results in lower correlations and less variability in concentration ratios, which is consistent with previous studies that demonstrated the importance of a good source representation (Kim et al., 2014; Korsakissok and Mallet, 2010b; Touma et al., 2006). Besides the ability to include detailed source characteristics and to use observed meteorology, the OPS model has some additional advantages. We have shown that looking at individual stacks can provide valuable information about the underlying transport and dispersion processes and how they are affected by source characteristics. Additionally, receptor sites can be positioned anywhere, which allows us to study the spatial variations at much higher resolution than currently possible with WRF-Chem.

At Cabauw, the difference between WRF-Chem and the WRF+OPS-point simulation is small, although the model-data mismatch at Cabauw is further reduced when observed meteorology is used. This leads to the question for which spatial extent a plume model is beneficial. In previous plume-in-grid models at high



resolution (<25 km) plumes or puffs are often injected in the Eulerian parent model when the width of the plume is similar to the grid size (Karamchandani et al., 2006; Kim et al., 2014; Korsakissok and Mallet, 2010b). According to the definition of the lateral dispersion factor in OPS this would mean that a plume will have

reached a horizontal width of 4 km (the resolution of the domain in which Cabauw is located) after about 8 km travel distance under well-mixed conditions. To test this, we compared a monthly average WRF-Chem $CO_2$ mixing ratio field in and around Rijnmond with a monthly averaged gridded OPS mixing ratio field. The OPS model was only applied for emissions within the Rijnmond area and therefore the distance outside the WRF-Chem domain 4 at which the mixing ratio fields become similar gives an indication of the spatial extent for

which the OPS model is still beneficial. We find that the difference between the mixing ratio fields disappears quickly outside the Rijnmond area and WRF-Chem and OPS become similar at about 10–14 km outside the boundary of domain 4.

## 5 Conclusions

Our ultimate ambition is to quantify the total urban $CO_2$ budget using multiple observation sites and an inverse

modelling system. Such information could be used to monitor the impact of implemented policies and progress towards objectives. Based on the work reported here, we state that the modelling framework should ideally consist of a Eulerian model in combination with a plume model for point source emissions within the city, preferably driven by locally observed meteorology. The use of a plume model is inevitable to correctly represent the transport of point source emissions in a diameter closer than ~10 km to the site. Although the additional

computational demand with the OPS plume model is limited, detailed model input is required given that the results are very sensitive to source characteristics and wind fields. Given the importance of observed local meteorology for the model performance, we strongly recommend inclusion of a (simple) meteorological station in any similar monitoring set-up. Also, Lagrangian particle dispersion models driven by WRF meteorological fields have proven useful in describing the transport of point source emissions and in inverse modelling (Brioude

et al., 2013; Pan et al., 2014; Srinivas et al., 2016), but such set-up would suffer from wind field errors. The optimal set-up for an urban monitoring network requires a semi-urban measurement site (here ~30 km from the urban area with no other urban areas in between) and at least one additional urban measurement site (here at the edge of the urban area, at ~7 km from the city centre). The semi-urban site provides a robust and integral constraint on the urban fluxes and can be used in combination with a high-resolution Eulerian model framework.

The urban measurement site can provide useful information about local differences, such as the dominance of road traffic in a certain source area or local changes due to implemented measures. Observing additional species besides CO, like $^{14}CO_2$, $^{13}CO_2$, $O_2/N_2$, $NO_2$, $SO_2$ or black carbon, could be a useful extension of our framework for identifying source sector contributions. Such a set-up is a promising step towards independent verification of urban $CO_2$ budgets.

**Data availability**

Observations from Zweth and Westmaas and the TNO-MACC III emission inventory are available via TNO (hugo.deniervandergon@tno.nl). Lutjewad and Cabauw observations can be downloaded from the



GLOBALVIEWplus product (Cooperative Global Atmospheric Data Integration Project, 2015). The Dutch Emission Registration emission inventory can be accessed online (http://www.emissieregistratie.nl/).

**Acknowledgements**

This research was partly funded by EIT Climate-KIC project Carbocount-CITY (APIN0029_2015-3.1-029_P040-04) and the EIT Climate KIC Fellows programme (ARED0004_2013-1.1-008_P017-0x). We sincerely thank the RIVM LML (Westmaas) and Hoogheemraadschap Delfland (De Zweth) for allowing us access to and the use of facilities on their sites. We thank Ferd Sauter (RIVM) for helping us with setting up the
OPS model for this study and for correcting the manuscript. Finally, we acknowledge ECN for providing us the Cabauw $CO_2$ observations, Huilin Chen for giving us access to the Lutjewad observations, and the Dutch Emission Registration for providing the point source characteristics needed for OPS.

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



**Table 1: Overview of SNAP categories and the vertical distribution of point source emissions in WRF-Chem.**

| SNAP | Description | % of point source emissions per model layer [m above surface] | | | | |
|---|---|---|---|---|---|---|
| | | 0–55 m | 55–130 m | 130–235 m | 235–360 m | >360 m |
| **1** | Combustion in energy and transformation industries | | | 18.5 % | 42 % | 39.5 % |
| **2** | Non-industrial combustion plants | | | | | |
| **3** | Combustion in manufacturing industry | 12.2 % | 37.3 % | 46.2 % | 4.3 % | |
| **4** | Production processes | 12. 2 % | 37.3 % | 46.2 % | 4.3 % | |
| **5** | Extraction and distribution of fossil fuels | | | | | |
| **6** | Solvents and other product use | | | | | |
| **7** | Road transport | | | | | |
| **8** | Other mobile sources and machinery | 100 % | | | | |
| **9** | Waste treatment and disposal | | 16.5 % | 44.5 % | 39 % | |
| **10** | Agriculture | | | | | |

**Table 2: Overview of the simulations, which model is used to calculate the urban plume mixing ratio from point sources in the Rijnmond area, how point sources are represented and the source of meteorological conditions.**

| Simulation name | Point source contribution | Point source representation | Meteorological input |
|---|---|---|---|
| **WRF-Chem** | WRF-Chem | area | WRF-Chem |
| **WRF+OPS-area** | OPS | area | WRF-Chem |
| **WRF+OPS-point** | OPS | point | WRF-Chem |
| **WRF+OPS-point-obsmet** | OPS | point | observations |


**Table 3: Statistics for WRF-Chem daytime (8:00–17:00 LT) average meteorological variables and total $CO_2$ and CO mixing ratios as compared to observed daytime averages (full simulation period). $\overline{X_{obs}}$ is the average observed mixing ratio and N gives the number of days included. This table shows that WRF-Chem is able to represent day-to-day variations in meteorological conditions and mixing ratios, except for the wind direction.**

| Variable | Site | $R^2$ | RMSE | bias | $\overline{X_{obs}}$ | N |
|---|---|---|---|---|---|---|
| **Temperature** | Rotterdam airport | 0.77 | 2.5 °C | + 0.9 °C | | 90 |
| **Specific humidity** | Rotterdam airport | 0.81 | 1.0 g kg$^{-1}$ | + 0.5 g kg$^{-1}$ | | 90 |
| **Wind speed** | Rotterdam airport | 0.72 | 1.2 m s$^{-1}$ | <0.1 m s$^{-1}$ | | 90 |
| **Wind direction** | Rotterdam airport | 0.20 | 53 degrees | - 13 degrees | | 90 |
| **$CO_2$ mixing ratio** | Westmaas | 0.65 | 8.8 ppm | + 1.1 ppm | 418 ppm | 83 |
| | Zweth | 0.45 | 13.0 ppm | + 2.5 ppm | 423 ppm | 85 |
| | Cabauw (60 m) | 0.48 | 10.6 ppm | + 3.6 ppm | 417 ppm | 86 |
| **CO mixing ratio** | Westmaas | 0.53 | 55 ppb | - 23 ppb | 187 ppb | 83 |
| | Zweth | 0.41 | 69 ppb | - 1 ppb | 198 ppb | 85 |
| | Cabauw (60 m) | 0.35 | 53 ppb | + 18 ppb | 156 ppb | 89 |



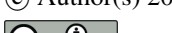



**Table 4: Statistics for the distribution of the observed and modelled (WRF-Chem) urban plume mixing ratios ($\Delta CO_2$ and $\Delta CO$) at the Zweth and Cabauw site. N is number of hours included for either the observed or simulated time series. The $R^2$ in the final column is based on co-sampling of WRF-Chem with the observations. The agreement between WRF-Chem and the observations is satisfactory when considering the distribution of the plume mixing ratios, but the low explained variance when co-sampling suggests a large impact of transport errors on individual plumes.**

| Species | Site | Obs/model | Median | 80th percentile | N | $R^2$ |
|---|---|---|---|---|---|---|
| $CO_2$ | Zweth | Observed | 9.7 ppm | 17.3 ppm | 284 | |
| | | WRF-Chem | 8.7 ppm | 17.2 ppm | 250 | 0.05 |
| | Cabauw (60 m) | Observed | 6.0 ppm | 9.1 ppm | 32 | |
| | | WRF-Chem | 5.6 ppm | 6.7 ppm | 36 | <0.01 |
| $CO$ | Zweth | Observed | 29 ppb | 57 ppb | 274 | |
| | | WRF-Chem | 20 ppb | 49 ppb | 208 | 0.01 |
| | Cabauw (60 m) | Observed | 13 ppb | 28 ppb | 58 | |
| | | WRF-Chem | 17 ppb | 32 ppb | 53 | <0.01 |

**Table 5: Statistics for $CO_2$ point source peaks at Zweth in four different model simulations as compared to observations. N is number of hours included and the slope is based on a linear regression. $\Delta CO:\Delta CO_2$ denotes the mean (± 1σ standard deviation) of the urban plume concentration ratio in ppb ppm$^{-1}$.**

| Model run | $R^2$ | $\Delta CO:\Delta CO_2$ | $\Delta CO_2$ slope | N |
|---|---|---|---|---|
| **WRF-Chem** | 0.30 | 0.9 (±1.5) | 0.82 | 42 |
| **WRF+OPS-area** | 0.37 | 1.2 (±1.1) | 0.87 | 42 |
| **WRF+OPS-point** | 0.42 | 1.2 (±1.6) | 0.86 | 42 |
| **WRF+OPS-point-obsmet** | 0.52 | 0.7 (±0.6) | 0.99 | 40 |
| | | | | |
| *Observed* | | *0.7 (±0.4)* | | |



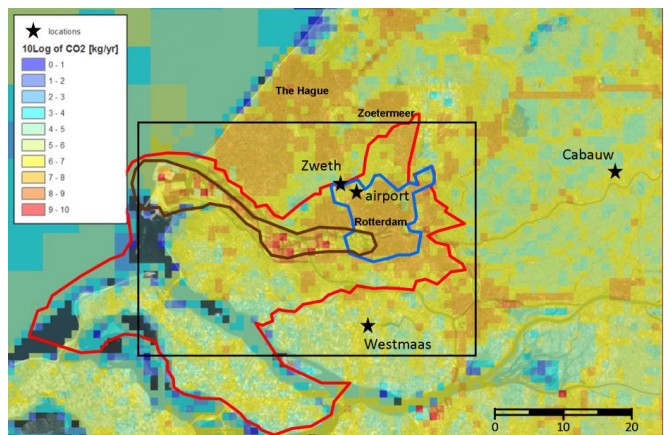

900 **Figure 1: CO₂ emission map of the Rijnmond area (red outline), including the city of Rotterdam (blue outline) and the port area (brown outline); the observation sites are indicated with black stars (Lutjewad is shown in Fig. 2). The boundaries of domain 4 in WRF-Chem are indicated by the black square. Source: Netherlands PRTR (2014).**

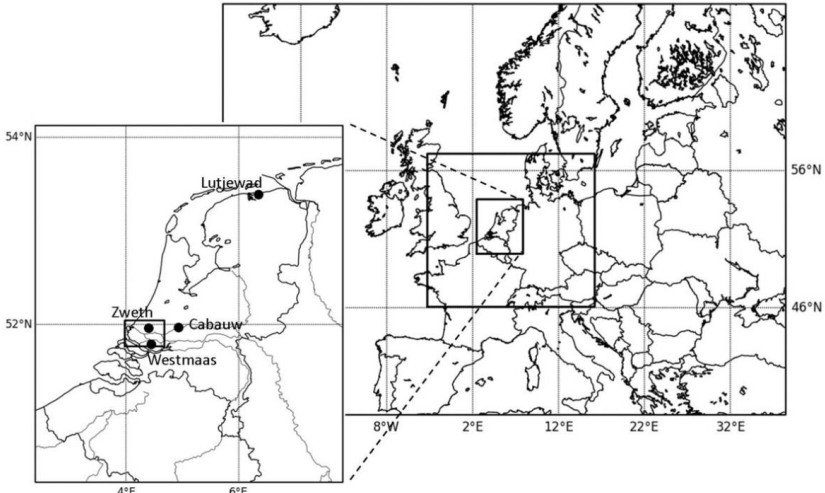

**Figure 2: Location of the domains is indicated with squares. The horizontal resolutions of the domains are (from outer to inner domain): 48x48 km, 12x12 km, 4x4 km and 1x1 km. Black circles represent the observation sites.**





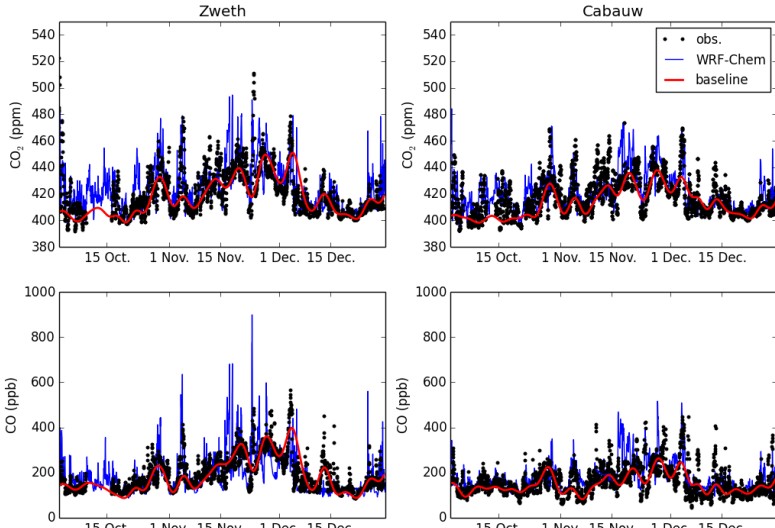

**Figure 3:** Time series of modelled (WRF-Chem) and observed $CO_2$ and CO mixing ratios at Zweth (left) and Cabauw (right). The observation-based baseline used in this study is also shown.

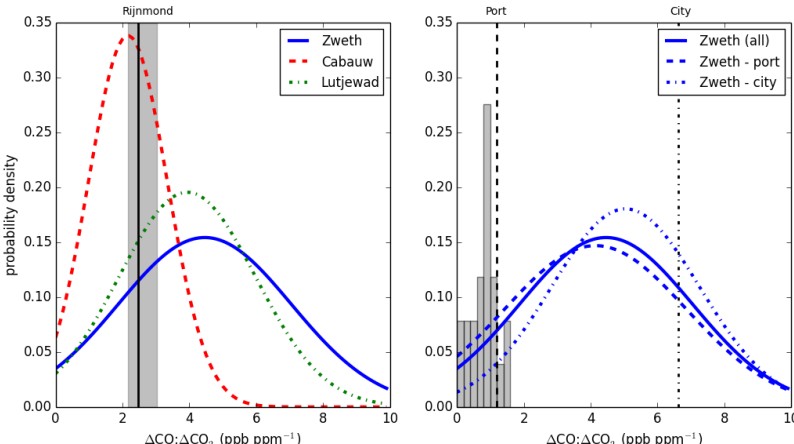

**Figure 4:** Left: Smooth Gaussian fit of probability density functions of observed $\Delta CO:\Delta CO_2$ at the Zweth, Cabauw and Lutjewad measurement sites. The solid vertical line (shaded area) shows the mean emission ratio (Q1–Q3 range) for all emissions integrated over the Rijnmond area (see Fig. 1). Right: The Zweth observations separated into two distinct source areas based on the observed wind direction. The dash-dotted and dashed vertical lines represent the mean emission ratios from the residential area and the port, respectively. Generally, there is a reasonable match between the bottom-up emission ratio and the concentration-derived ratio, but observed ratios from the Zweth-port wind sector are much higher than expected because of the intermittency of plume transport from the many stacks in this area. The grey bars in the right panel show the point source events selected in Sect. 3.3.




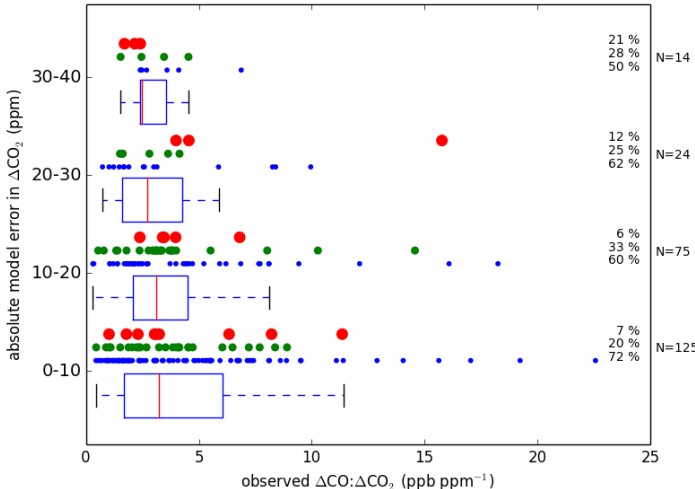

**Figure 5:** This figure shows four classes of the absolute model error in $\Delta CO_2$ compared with the Zweth measurement
site. For each class two quantities are displayed. 1) A whisker plot of observed $\Delta CO:\Delta CO_2$, which shows that the
largest absolute $\Delta CO_2$ model error (y-axis) is related to small observed concentration ratios (x-axis). This indicates an
important role for low-ratio stack emissions (industrial and power plant sources) in the large model error class. 2) A
coloured scatter plot for which data points are divided into three classes based on the absolute error in simulated
wind direction (<20 degrees in small blue dots on bottom row, 20–40 degrees in larger green dots on middle row, and
>40 degrees in large red dots on top row). Each dot represents one hour. The percentage contribution of each wind
direction error class to the total number of data points (N) is shown on the right. These numbers show that the model
error in wind direction also plays an important role in the $\Delta CO_2$ model error.

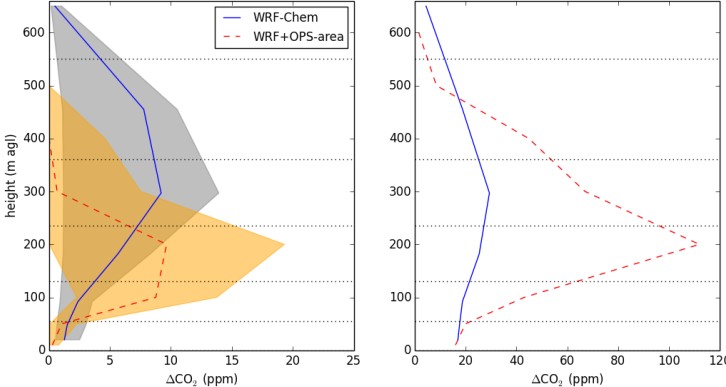

**Figure 6:** Vertical profiles of the median (Q1–Q3) (left panel) and maximum (right panel) $\Delta CO_2$ mixing ratio at 14 h
UTC at about 500 m from an energy production point source in WRF-Chem and WRF+OPS-area. The horizontal
lines represent the boundaries of the vertical levels in WRF-Chem. Emissions are taking place in levels 3, 4 and 5 in
WRF-Chem or at 130, 235 and 360 m in WRF+OPS-area. The figure shows on average lower mixing ratios at ground
level in WRF+OPS-area than in WRF-Chem, despite an identical treatment of the vertical emission structure.
WRF+OPS-area also shows higher maximum values, reflecting a reduction in vertical dispersion compared to the
Eulerian box representation in WRF-Chem.





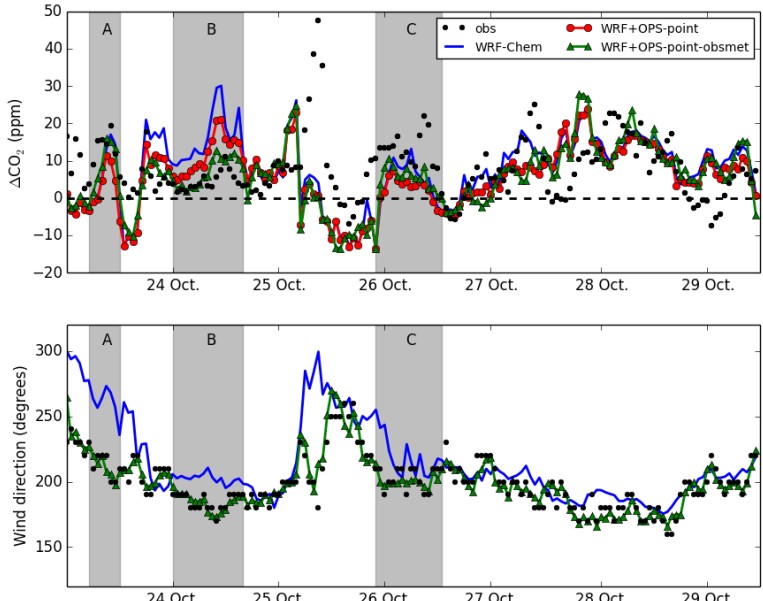

**Figure 7: Time series of ΔCO₂ at Zweth from observations and three model simulations (top panel) and of the wind direction at Rotterdam airport from WRF-Chem, WRF+OPS-point-obsmet, and observations (bottom panel). Shaded areas indicate specific events discussed in more detail in the text.**

940