# Peer review of "A multi-model approach to monitor emissions of CO2 and CO from an urban-industrial complex"

_Atmospheric Chemistry and Physics, 2017_

## Referee Comment (RC1) · J. Turnbull (Referee) · 21 Jul 2017

This paper describes a combined Eulerian/plume model approach to evaluate CO2 (and CO) emissions, using Rotterdam, The Netherlands as an example. The authors show clearly that embedding a plume model within the Eulerian model improves the overall model fidelity in areas close to point sources. The results demonstrate that this is due to the Eulerian model resolution being insufficient to capture the details of nearby plumes. Presumably with infinite computing power, the Eulerian model could overcome this limitation, but embedding the plume model is a more computationally efficient solution. Further away from the point sources, the plume model doesn't add much, since the Eulerian resolution becomes sufficient at these spatial scales. They also evaluate the effect of wind direction biases in the Eulerian model (WRF-Chem is

used in this case), and show that using observed meteorology makes a big improvement in the model fidelity close to point sources – this is not a surprise, but nonetheless is a nice result.

This is a well-written, clear and easy to read paper. It is entirely appropriate for publication in ACP and I recommend it be accepted with minor modifications, detailed below.

General comments: 1. Is there a reason why there are no runs where WRF-Chem is nudged with the local meteorology? Given the results that show using local meteorology in the plume model really helps, this would be an obvious test to do. 2. I would like to see a bit of discussion about the relative difficulty of running the combined Eulerian/plume model simulations. How much more computation time is needed vs running WRF-Chem alone? How much additional effort (not computation time, but people time) is required – is it set up to run easily, or does someone have to sit there and do each plume run individually? 3. The method presented here requires that the point sources are known from a bottom-up inventory and that the bottom-up information has the correct locations. That's going to be difficult in many cities where the information simply isn't available. Some comment on this (in the conclusions or discussion) is needed.

Specific comments: Line 33 (abstract). "Inevitable" seems a strong word to use – clearly the plume model helps a lot, but one could imagine other ways to address the same problem. Tone down the words used. Same for line 558 in conclusions.

Line 65. Consider revising wording from ". . .that mask the urban signal" to "that mask the overall urban signal".

Lines 175-189 – CO fluxes. I agree that ignoring oxidation of hydrocarbons is probably reasonable for the winter months considered here, but I suspect that biofuel combustion might be important. Biofuel combustion (such as wood fires for home heating) tends to be quite inefficient with high $CO:CO_2$ ratios, so that even a small contribution to the $CO_2$ source might mean a significant CO source. The $CO_{biofuel}$ source could be estimated by combining the $CO_{2biofuel}$ flux estimate with an estimate of the emission

ratio. Andreae and Merlet 2001 is a good (even if old) resource to make some guesses about the emission ratio. Andreae, M.O., Merlet, P., 2001. Emission of trace gases and aerosols from biomass burning. Global Biogeochemical Cycles 15, 955-966.

Lines 196-200. I take it the plume model is run forward (not backward as is common when plume models are run as a stand-alone)? Consider stating this explicitly to clarify.

Lines 280-291. The choice of background definition from the baseline values is clearly more workable in this particular environment than using an upwind background. Nonetheless, it could be a problem on occasions when the incoming air is unusually polluted – the baseline background will not account for that. Please add a comment on this.

Lines 333-347. I agree with the interpretation as described here, but I think you also need to discuss other possible explanations for the difference between observed and modeled $CO:CO_2$ ratios, and why these possibilities are less likely. Is it possible that the point source $CO:CO_2$ ratio is in fact higher than reported? Perhaps the inventories are wrong, and/or the industries are not scrubbing CO as effectively as they claim to? CO from biofuel is not included in the model (see also earlier comments) – how would including this alter the modelled ratios?

Line 357. I don't think you ever spell out what RMSE is. Please do so the first time you use it.

Lines 365-367. This effect has been seen before. Please add appropriate references.

Line 368. I am not sure what you mean by "there is no co-sampling for this comparison". Please revise for clarity.

Lines 369-370. You remove low wind speed data. Some additional discussion about the overall performance of the model when all data is included is needed. Do you conclude that it is generally difficult to model low wind speed time periods and they should always be discarded? In many environments, low wind speeds are when it is

particularly cold and more CO2 is generated for heating, so removing this data might bias the overall analysis to lower emissions.

Lines 374-376. Sentence beginning "However, if we co-sample...". I don't understand how this is different than the previous analysis you discuss. Please clarify.

Lines 393 – 395. "At Cabauw..." This sentence seems out of place.

Section 3.3. I would like to see some plots of the comparison in addition to the summary in the table. Perhaps as supplementary material?

Lines 415-417. Clarify that you don't show the data for this particular test.

Line 470. "specifically constrain", not "constrain specifically".

Jocelyn Turnbull, July 21 2017
* * *

---

## Referee Comment (RC2) · Anonymous Referee #2 · 29 Aug 2017

Summary/General comments:

Super et al. combine observations of CO2 and CO from urban and ex-urban sites in the Netherlands with an Eulerian modeling scheme (WRF-Chem) that explicitly accounts for plumes for large point sources to evaluate the utility of different urban/exurban observations and determine the utility of an Eulerian model in quantifying urban fluxes of CO2. This is a thorough, well written paper that contributes significantly to the field of urban GHG research and is well placed in ACP. I enthusiastically recommend publication once these minor comments have been addressed.

Major comments: The largest critique is the breadth of the conclusions implied in the abstract. Most pointedly, line 25, should instead state a plume model can be added to the model framework to account for point sources – the authors have shown that in

an Eulerian model of typical regional resolution plumes an incorrectly represented and a plume model can fix this. However, a Lagrangian model, LES model, or very high resolution Eulerian model may not require this and the authors have not demonstrated as such. Similarly, line 33-34 are overstated. Integration of a plume model is not inevitable, as the authors have not shown alternatives are inadequate. The authors have shown that integration of a plume model is a possible solution for using a regional lagrangian model and surface point observations for CO2.

The authors have shown in compelling fashion the need for accounting for stack CO2 emissions w/ a plume framework. It is interesting that this is not the case for CO, and it would be nice for that to be highlighted. Further, I wonder then if a plume model representation would be important for methane? Also, the authors are considering surface, point observations. If total column observations are considered, is a plume model essential or is the vertical dilution now irrelevant? This is perhaps a question beyond the current analysis, but it would be an interesting point to comment on.

Detailed comments:

Line 57: This is dependent on urban typology and emission characteristics. The authors should acknowledge this limitation here.

Lines 86-90: Other cities have also been studies – most notably Boston and Indianapolis, there are a sequence of INFLUX papers that it would be appropriate to cite here.

Line 175-183: I worry about this sweeping the VOC CO production under the rug. How much does this really matter? I suspect the authors' analysis is robust to this as the VOC CO production is embedded within the determination of the boundary condition, and thus ignoring it is ok as the amount produced in the near field (within 24 hours) is modest. I'd like a little more discussion of this, and estimates of how much this may matter if the same approach is taken in the summer?

Title: I'd suggest a change as the manuscript is really not monitoring CO emissions,

but leveraging CO to better interpret CO2 emissions, and the current title is a little misleading.

---

## Author Comment (AC1) · 19 Sep 2017

We would like to thank the reviewers for their enthusiasm about our study and for the comments on our work. The review comments have been helpful in reflecting on our work and pointing out parts that required further improvements. Below we address specific issues mentioned by the reviewers point by point. The manuscript has been updated accordingly (changes are highlighted).

Reviewer #1 (J. Turnbull) This paper describes a combined Eulerian/plume model approach to evaluate CO2 (and CO) emissions, using Rotterdam, The Netherlands as an example. The authors show clearly that embedding a plume model within the Eulerian model improves the overall model fidelity in areas close to point sources. The

results demonstrate that this is due to the Eulerian model resolution being insufficient to capture the details of nearby plumes. Presumably with infinite computing power, the Eulerian model could overcome this limitation, but embedding the plume model is a more computationally efficient solution. Further away from the point sources, the plume model doesn't add much, since the Eulerian resolution becomes sufficient at these spatial scales. They also evaluate the effect of wind direction biases in the Eulerian model (WRF-Chem is used in this case), and show that using observed meteorology makes a big improvement in the model fidelity close to point sources – this is not a surprise, but nonetheless is a nice result. This is a well-written, clear and easy to read paper. It is entirely appropriate for publication in ACP and I recommend it be accepted with minor modifications, detailed below.

General comments: 1. Is there a reason why there are no runs where WRF-Chem is nudged with the local meteorology? Given the results that show using local meteorology in the plume model really helps, this would be an obvious test to do.

We sincerely thank the reviewer for this comment and acknowledge that nudging local meteorological data would be a good step forward. However, we tested the WRF-FDDA system for a short period and found no consistent improvement, as periods of better wind field representation are alternated with periods with decreased performance. Moreover, previous studies have shown that the results from nudging are highly dependent on the type of data that is nudged and the grid resolution (1,2). Therefore, we believe that more time needs to be dedicated to understanding the effects of wind field nudging to improve the WRF wind field representation. We plan to do so in a next study and thus decided to mention data assimilation as a point for future research.

1 Deng, A., and N. Seaman, G. Hunter, and D. Stauffer, 2004: Evaluation of interregional transport using the MM5-SCIPUFF system. J. Appl. Meteor., 43, 1864–1886. 2 Deng, A., and D. Stauffer, 2006: On improving 4-km mesoscale model simulations. J. Appl. Meteor., 45, 361–381.

2. I would like to see a bit of discussion about the relative difficulty of running the combined Eulerian/plume model simulations. How much more computation time is needed vs running WRF-Chem alone? How much additional effort (not computation time, but people time) is required – is it set up to run easily, or does someone have to sit there and do each plume run individually?

The OPS plume model is a very easy and fast model to run. For the $\sim$100 point source emissions in our domain it needs about 20-25 seconds to calculate hourly mixing ratios at 4 sites for the full three months. In contrast, with the current set-up WRF-Chem needs several weeks to simulate the three months using 8 cores. Since OPS can handle multiple sources and receptor sites at once, it only needs to be run once for each species. The OPS model only needs an emission file and a file with general information about the run (e.g. the period). However, this doesn't allow us to separate between source types; i.e. we do not know which part of the signal is from industrial sources and which part from energy production sites. Nevertheless, this could be done by doing separate simulations per source type with limited additional effort. Thus the combined modelling system doesn't require much extra effort compared to running WRF-Chem alone. This is now shortly mentioned in lines 532-534.

3. The method presented here requires that the point sources are known from a bottom-up inventory and that the bottom-up information has the correct locations. That's going to be difficult in many cities where the information simply isn't available. Some comment on this (in the conclusions or discussion) is needed.

We agree with the reviewer that such detailed source characteristics benefit the model results. Except for the location (which could be inferred form Google Maps for example), such details are probably not available everywhere. We added some discussion on this in lines 545-548.

Specific comments: Line 33 (abstract). "Inevitable" seems a strong word to use – clearly the plume model helps a lot, but one could imagine other ways to address the

same problem. Tone down the words used. Same for line 558 in conclusions.

We thank the reviewer for the suggestion and have revised the text to "adds substantially to" in line 32 and to "is of great added value" in line 581.

Line 65. Consider revising wording from "... that mask the urban signal" to "that mask the overall urban signal".

We have revised the text to "that mask the overall urban signal" in line 63.

Lines 175-189 – CO fluxes. I agree that ignoring oxidation of hydrocarbons is probably reasonable for the winter months considered here, but I suspect that biofuel combustion might be important. Biofuel combustion (such as wood fires for home heating) tends to be quite inefficient with high CO:CO2 ratios, so that even a small contribution to the CO2 source might mean a significant CO source. The CObiofuel source could be estimated by combining the CO2biofuel flux estimate with an estimate of the emission ratio. Andreae and Merlet 2001 is a good (even if old) resource to make some guesses about the emission ratio. Andreae, M.O., Merlet, P., 2001. Emission of trace gases and aerosols from biomass burning. Global Biogeochemical Cycles 15, 955-966.

We thank the reviewer for pointing out the importance of biofuel CO emissions. Indeed, biofuel combustion is an important source of CO, also in our study domain. In fact, the biofuel emissions - such as from wood stoves or biomass plants - are already part of the emission data used in our study, although they are part of the total emissions and cannot be quantified separately. The CO and CO2 biofuel emissions thus do not have to be added in the model, but we do need to mention the term in Eq. 2 and the subsequent descriptions. We apologize for this omission in our manuscript. We have now included the biofuel terms in Eq. 2 and have added biofuel emissions in lines 223-225.

Lines 196-200. I take it the plume model is run forward (not backward as is common when plume models are run as a stand-alone)? Consider stating this explicitly to clarify.

We have rewritten line 201 to "The model keeps track of a trajectory forward in time [...]"

Lines 280-291. The choice of background definition from the baseline values is clearly more workable in this particular environment than using an upwind background. Nonetheless, it could be a problem on occasions when the incoming air is unusually polluted – the baseline background will not account for that. Please add a comment on this.

Generally our baseline slightly overestimates the background, meaning that the $\Delta CO2$ and $\Delta CO$ mixing ratios are really local additions. However, the reviewer is correct that for short periods of high pollution our background method underestimates the background mixing ratio. This is now mentioned in lines 293-294.

Lines 333-347. I agree with the interpretation as described here, but I think you also need to discuss other possible explanations for the difference between observed and modeled CO:CO2 ratios, and why these possibilities are less likely. Is it possible that the point source CO:CO2 ratio is in fact higher than reported? Perhaps the inventories are wrong, and/or the industries are not scrubbing CO as effectively as they claim to? CO from biofuel is not included in the model (see also earlier comments) – how would including this alter the modelled ratios?

We thank the reviewer for the suggested alternative explanations. There is indeed an uncertainty in the reported stack emissions. Nevertheless, these figures are thoroughly quality-checked by the Netherlands Environmental Assessment Agency according to the IPCC guidelines, as it is part of the National Inventory Report. Therefore, these figures are relatively accurate and it is unlikely that this is the (main) cause for the model-data mismatch. We added this discussion in lines 349-351. As mentioned before, biofuels are included after all. Another potential explanation is that the CO:CO2 emission ratios are not constant in time, while they are kept constant in the model by applying similar temporal profiles for CO and CO2. We added this suggestion in lines

351-353.

Line 357. I don't think you ever spell out what RMSE is. Please do so the first time you use it.

RMSE has been spelled out now in line 364.

Lines 365-367. This effect has been seen before. Please add appropriate references.

We have added a sentence saying this, including references, in line 368.

Line 368. I am not sure what you mean by "there is no co-sampling for this comparison". Please revise for clarity.

Here, we compare two data sets. For the first one, we take all data that is above the baseline and satisfies our criteria, separately for the observed and modelled time series. For the second one, we make a similar selection for the observations, but then co-sample the modelled time series. So, whereas the two data sets for the first comparison can have a different size and include different times, for the second comparison they have equal size and contain the same times. We have tried to clarify this in lines 376-378.

Lines 369-370. You remove low wind speed data. Some additional discussion about the overall performance of the model when all data is included is needed. Do you conclude that it is generally difficult to model low wind speed time periods and they should always be discarded? In many environments, low wind speeds are when it is particularly cold and more CO2 is generated for heating, so removing this data might bias the overall analysis to lower emissions.

Data analysis shows that for the results in Table 4 the addition of low wind speed data slightly increases the median and percentile values in a similar way for the observations and the WRF output. Thus the removal of low wind speed data has limited impact on the results, which is now mentioned in lines 379. However, for the subset in Table 5 the inclusion of low wind speed data (in total 3 data points) significantly deteriorates

the results. This indicates that the models have difficulty correctly representing such stagnant conditions, which is now mentioned in lines 414-416.

Indeed, removal of low wind speed data can cause a bias in the estimated emissions, which has now been mentioned in lines 516-517. Similarly, the removal of night time data also causes a bias by not taking into account hours with usually lower emissions. However, including stable stratified and stagnant conditions with a large model bias will result in a large posterior uncertainty and most studies exclude these data (3-5).

3 Boon, A., Broquet, G., Clifford, D. J., Chevallier, F., Butterfield, D. M., Pison, I., Ramonet, M., Paris, J. D., and Ciais, P.: Analysis of the potential of near-ground measurements of CO2 and CH4 in London, UK, for the monitoring of city-scale emissions using an atmospheric transport model, Atmos. Chem. Phys., 16, 6735-6756, 10.5194/acp-16-6735-2016, 2016. 4 Bréon, F. M., Broquet, G., Puygrenier, V., Chevallier, F., Xueref-Remy, I., Ramonet, M., Dieudonné, E., Lopez, M., Schmidt, M., Perrussel, O., and Ciais, P.: An attempt at estimating Paris area CO2 emissions from atmospheric concentration measurements, Atmos. Chem. Phys., 15, 1707-1724, 10.5194/acp-15-1707-2015, 2015. 5 Lauvaux, T., Miles, N. L., Deng, A., Richardson, S. J., Cambaliza, M. O., Davis, K. J., Gaudet, B., Gurney, K. R., Huang, J., O'Keefe, D., Song, Y., Karion, A., Oda, T., Patarasuk, R., Razlivanov, I., Sarmiento, D., Shepson, P., Sweeney, C., Turnbull, J., and Wu, K.: High-resolution atmospheric inversion of urban CO2 emissions during the dormant season of the Indianapolis Flux Experiment (INFLUX), J. Geophys. Res.-Atmos., 121, 5213-5236, 10.1002/2015jd024473, 2016.

Lines 374-376. Sentence beginning "However, if we co-sample...". I don't understand how this is different than the previous analysis you discuss. Please clarify.

See our previous comment on the difference between the two analyses. This has been clarified in lines 384-386.

Lines 393 – 395. "At Cabauw ..." This sentence seems out of place.

Previously, the analyses have been done for both Zweth and Cabauw. However, we find that the impact of stack emissions at the Cabauw site is so limited that we don't apply the plume model to this location. That is what is stated in lines 403-404.

Section 3.3. I would like to see some plots of the comparison in addition to the summary in the table. Perhaps as supplementary material?

We have added the following figure to an appendix. It shows the results related to the ratios (left panel) and to the $\Delta CO2$ slope (right panel).

Figure A1: Left: A scatter plot of $\Delta CO$ and $\Delta CO2$, where the slopes (represented by lines) represent the $\Delta CO:\Delta CO2$ ratio for the observed and modelled values. The slope of WRF+OPS-point-obsmet coincides with the slope of the observations, suggesting a good agreement. Right: A scatter plot of simulated $\Delta CO2$ to observed $\Delta CO2$. The slope of WRF+OPS-point-obsmet coincides with the 1:1 line (dotted line), suggesting a good agreement with the observations.

Lines 415-417. Clarify that you don't show the data for this particular test.

We have clarified this in line 429.

Line 470. "specifically constrain", not "constrain specifically".

We have revised this in line 484.

Reviewer #2 Summary/General comments: Super et al. combine observations of CO2 and CO from urban and ex-urban sites in the Netherlands with an Eulerian modeling scheme (WRF-Chem) that explicitly accounts for plumes for large point sources to evaluate the utility of different urban/exurban observations and determine the utility of an Eulerian model in quantifying urban fluxes of CO2. This is a thorough, well written paper that contributes significantly to the field of urban GHG research and is well placed in ACP. I enthusiastically recommend publication once these minor comments have been addressed.

Major comments: The largest critique is the breadth of the conclusions implied in the abstract. Most pointedly, line 25, should instead state a plume model can be added to the model framework to account for point sources – the authors have shown that in an Eulerian model of typical regional resolution plumes an incorrectly represented and a plume model can fix this. However, a Lagrangian model, LES model, or very high resolution Eulerian model may not require this and the authors have not demonstrated as such. Similarly, line 33-34 are overstated. Integration of a plume model is not inevitable, as the authors have not shown alternatives are inadequate. The authors have shown that integration of a plume model is a possible solution for using a regional lagrangian model and surface point observations for CO2.

We sincerely thank the reviewer for this comment and we agree that other solutions might be possible, such as LES or full Lagrangian models. Our intention was to stress that a Eulerian model alone at the current resolution is not sufficient to represent point source emissions, and that a plume model can overcome some of the limitations. The use of a multi-model framework or a different type of model will depend on the application, and the resolution and scale. As we have shown, the plume model only has an impact up to about 10-15 km from a source. For other models, this might be different. Also, the models differ in how much effort is needed to set it up and do simulations, which might also be an important consideration. In our work we have demonstrated the use of an easy-to-use plume model, but we do not want to rule out other options. Therefore, we have revised the text to "[...], adding a plume model to the model framework is beneficial [...]" in line 25 and "adds substantially to" in line 32.

The authors have shown in compelling fashion the need for accounting for stack CO2 emissions w/ a plume framework. It is interesting that this is not the case for CO, and it would be nice for that to be highlighted. Further, I wonder then if a plume model representation would be important for methane? Also, the authors are considering surface, point observations. If total column observations are considered, is a plume model essential or is the vertical dilution now irrelevant? This is perhaps a question

beyond the current analysis, but it would be an interesting point to comment on.

We thank the reviewer for this interesting comment. Indeed, the plume model has limited added value for CO as most of the CO emissions are coming from area sources (stressed in lines 531-532). For methane the dominant source type in our domain is waste treatment and disposal, especially landfills. Depending on the size of the site relative to the size of the model grid, such sources could usually be considered area sources. Another important source of methane emissions is gas leakages. These are likely point sources, but the location of these leakages is unknown and it can be difficult to add these to a plume model.

Column integrals are of interest in the light of upcoming high-resolution missions. With column observations the vertical distribution would indeed become less relevant. However, the plume model also reduces the horizontal distribution of a point source compared to the grid box averaging done by a Eulerian model. In that sense, a plume model could still be useful. We plan to do more work on column observations in the near future. Whether a plume model is useful in that context would be an excellent question to pose for that work.

Detailed comments: Line 57: This is dependent on urban typology and emission characteristics. The authors should acknowledge this limitation here.

A note has been added to lines 499-500 to clarify that our conclusions are valid for the Rijnmond area and cannot be generalized to other areas without careful consideration of the urban typology.

Lines 86-90: Other cities have also been studies – most notably Boston and Indianapolis, there are a sequence of INFLUX papers that it would be appropriate to cite here.

We apologize to the reviewer for having missed these references. We have added a reference related to urban scale monitoring of fossil fuel CO2 emissions (6) and another reference related to a high-resolution inversion of urban CO2 emissions using

a Lagrangian Particle Dispersion Model driven by WRF meteorological fields (7) (see lines 88-89). We believe that these studies are strongly related to our research and are relevant for our introduction into the topic of urban CO2 monitoring.

6 Turnbull, J. C., Sweeney, C., Karion, A., Newberger, T., Lehman, S. J., Tans, P. P., Davis, K. J., Lauvaux, T., Miles, N. L., Richardson, S. J., Cambaliza, M. O., Shepson, P. B., Gurney, K., Patarasuk, R., and Razlivanov, I.: Toward quantification and source sector identification of fossil fuel CO2 emissions from an urban area: Results from the INFLUX experiment, J. Geophys. Res.-Atmos., 120, 292-312, 10.1002/2014jd022555, 2015. 7 Lauvaux, T., Miles, N. L., Deng, A., Richardson, S. J., Cambaliza, M. O., Davis, K. J., Gaudet, B., Gurney, K. R., Huang, J., O'Keefe, D., Song, Y., Karion, A., Oda, T., Patarasuk, R., Razlivanov, I., Sarmiento, D., Shepson, P., Sweeney, C., Turnbull, J., and Wu, K.: High-resolution atmospheric inversion of urban CO2 emissions during the dormant season of the Indianapolis Flux Experiment (INFLUX), J. Geophys. Res.-Atmos., 121, 5213-5236, 10.1002/2015jd024473, 2016.

Line 175-183: I worry about this sweeping the VOC CO production under the rug. How much does this really matter? I suspect the authors' analysis is robust to this as the VOC CO production is embedded within the determination of the boundary condition, and thus ignoring it is ok as the amount produced in the near field (within 24 hours) is modest. I'd like a little more discussion of this, and estimates of how much this may matter if the same approach is taken in the summer?

In a study by Griffin et al. (2007), referenced in the manuscript, the contribution of hydrocarbon oxidation to the total CO production rate was investigated for two areas during high-pollution episodes. They found contributions of 5% for an area with significant biogenic hydrocarbon production and of 1% for an area dominated by anthropogenic CO emissions. In contrast, Hudman et al. (2008) (also referenced in the manuscript) come up with an estimate of more than 60% for the eastern United States. The main difference is that the Griffin study uses smaller domains, allowing hydrocarbons to be transported out of the domain before it can be converted into CO. Moreover, the domain of Hudman et al. covers an area with substantial biogenic fluxes and the dominant source of CO is the oxidation of isoprene.

Given the small size of our domain and the dominance of primary CO emissions, we assume the impact of hydrocarbon oxidation will be closer to the estimate of Griffin et al. Additionally, both studies were performed during summer, whereas the conditions during our study are less favourable for photochemistry. We thus believe that neglecting hydrocarbon oxidation in our study will introduce a bias, albeit a small one. This has been discussed in lines 183-184.

The two studies discussed here show that several factors will affect the estimated impact of hydrocarbons. Therefore, it is very difficult to make an estimate of how large this impact will be in our domain for summer months. We therefore believe that giving an estimate, as suggested by the reviewer, might be misleading at this point.

Title: I'd suggest a change as the manuscript is really not monitoring CO emissions, but leveraging CO to better interpret CO2 emissions, and the current title is a little misleading.

We thank the reviewer for this suggestion. However, we believe that CO emission are fully coupled to the CO2 emissions in our study through the use of fixed emission factors. So any updated CO2 emissions automatically also lock the CO emissions. Yet, we do acknowledge that CO emissions were not the main target in this study.

Please also note the supplement to this comment:
https://www.atmos-chem-phys-discuss.net/acp-2017-431/acp-2017-431-AC1-supplement.pdf

[revised manuscript text omitted]